# Modulation of Human Dendritic Cell Functions by Phosphodiesterase-4 Inhibitors: Potential Relevance for the Treatment of Respiratory Diseases

**DOI:** 10.3390/pharmaceutics15092254

**Published:** 2023-08-31

**Authors:** Hoang Oanh Nguyen, Laura Tiberio, Fabrizio Facchinetti, Giulia Ripari, Valentina Violi, Gino Villetti, Valentina Salvi, Daniela Bosisio

**Affiliations:** 1ImmunoConcEpT, CNRS UMR 5164, University of Bordeaux, 33000 Bordeaux, France; hnguyen@immuconcept.org; 2Department of Molecular and Translational Medicine, University of Brescia, 25123 Brescia, Italy; laura.tiberio@unibs.it (L.T.); g.ripari@unibs.it (G.R.); v.violi@studenti.unibs.it (V.V.); 3Department of Experimental Pharmacology and Translational Science, Corporate Pre-Clinical R&D, Chiesi Farmaceutici S.p.A., 43122 Parma, Italy; f.facchinetti@chiesi.com (F.F.); g.villetti@chiesi.com (G.V.)

**Keywords:** cDC1, cDC2, pDC, monocyte-derived DC, tolerogenic DCs, CD80, CD86, IL-12, TNFα, Tanimilast

## Abstract

Inhibitors of phosphodiesterase-4 (PDE4) are small-molecule drugs that, by increasing the intracellular levels of cAMP in immune cells, elicit a broad spectrum of anti-inflammatory effects. As such, PDE4 inhibitors are actively studied as therapeutic options in a variety of human diseases characterized by an underlying inflammatory pathogenesis. Dendritic cells (DCs) are checkpoints of the inflammatory and immune responses, being responsible for both activation and dampening depending on their activation status. This review shows evidence that PDE4 inhibitors modulate inflammatory DC activation by decreasing the secretion of inflammatory and Th1/Th17-polarizing cytokines, although preserving the expression of costimulatory molecules and the CD4+ T cell-activating potential. In addition, DCs activated in the presence of PDE4 inhibitors induce a preferential Th2 skewing of effector T cells, retain the secretion of Th2-attracting chemokines and increase the production of T cell regulatory mediators, such as IDO1, TSP-1, VEGF-A and Amphiregulin. Finally, PDE4 inhibitors selectively induce the expression of the surface molecule CD141/Thrombomodulin/BDCA-3. The result of such fine-tuning is immunomodulatory DCs that are distinct from those induced by classical anti-inflammatory drugs, such as corticosteroids. The possible implications for the treatment of respiratory disorders (such as COPD, asthma and COVID-19) by PDE4 inhibitors will be discussed.

## 1. Introduction 

cAMP is a key modulator of inflammation whose intracellular levels are regulated through hydrolysis by various phosphodiesterases (PDEs) [1]. PDE4, in particular, is a family of four genes (PDE4A, PDE4B, PDE4C and PDE4D) encoding cAMP-specific enzymes sharing a highly conserved catalytic domain and prominently expressed in immune cells.

PDE4 inhibitors increase cAMP intracellular levels by inhibiting cAMP hydrolysis, thus eliciting a broad spectrum of anti-inflammatory effects in virtually all cells of the immune system [2]. Because of this, the inhibition of PDE4 enzymes has been clinically investigated as a therapeutic strategy in a variety of pathological settings, including respiratory, dermatological and immune diseases, as well as cognitive and affective disorders [3,4].

Dendritic cells (DCs) are a specialized population of antigen-presenting cells, serving as sentinels of innate immunity and key initiators of adaptive and anti-viral immune responses. Deregulated DC functions are one major pathogenetic determinant of virtually all types of diseases [5]. Thus, DCs represent crucial targets for immunomodulatory drugs. This present review will describe how PDE4 inhibitors affect the immune-activating properties of DCs, which display relevant cAMP-specific PDE4 enzymatic activity mainly mediated by PDE4A [6]. Additionally, we will describe how the immunomodulatory effects of PDE4 inhibitors impact the treatment of some major respiratory diseases, highlighting the possible implications of DC modulation in such conditions.

## 2. Overview of PDE4 Inhibitors

Three small-molecule PDE4 inhibitors are currently approved and marketed for the treatment of human diseases: Roflumilast for chronic obstructive pulmonary disease (COPD), crisaborole for atopic dermatitis and Apremilast for psoriatic arthritis [7]. Roflumilast, developed as an oral drug, is the first approved PDE4 inhibitor as well as the only PDE4 inhibitor approved for respiratory diseases [8]. Roflumilast was approved in the EU and the US in 2010 and 2011, respectively, for the oral treatment of severe COPD [9]. Roflumilast proved capable of reducing inflammatory-mediated processes and the frequency of exacerbations in COPD patients with a chronic bronchitis phenotype [8]. Moreover, the potential utility of Roflumilast in non-cystic fibrosis bronchiectasis, a neutrophilic airway disease presenting with mucus hypersecretion, is under investigation in clinical trials (https://classic.clinicaltrials.gov/ct2/show/NCT03988816, accessed on 1 July 2023). However, the use of Roflumilast is limited by class-related side effects, such as nausea, diarrhea, weight loss and abdominal pain, resulting in both substantial treatment discontinuation in clinical practice and withdrawal from clinical trials. This has prompted the search for PDE4 inhibitors to be administered by inhalation to reduce the systemic exposure (and thus optimize drug safety) and maximize the therapeutic effect in the lung [10]. Tanimilast (international non-proprietary name of CHF6001) is an example of a novel inhaled PDE4 inhibitor that has progressed to clinical studies in COPD and asthma [11]. Tanimilast showed anti-inflammatory properties in various inflammatory cells, including leukocytes derived from asthma and COPD patients, as well as in experimental rodent models of pulmonary inflammation [11,12,13]. Inhaled Tanimilast has reached phase III clinical development by showing promising pharmacodynamic results associated with a good tolerability and safety profile, with no evidence of PDE4 inhibitor class-related side effects [11]. Apremilast is currently under clinical evaluation as a therapy to mitigate the severe inflammatory reactions associated with COVID-19 pneumonia (https://classic.clinicaltrials.gov/ct2/show/NCT04590586, accessed on 1 July 2023). Additionally, a number of molecules are currently under clinical trial for the treatment of respiratory diseases, as recently reviewed in several works [4,7,14,15,16] and briefly summarized in Table 1.

Several clinical trials for the treatment of COPD and psoriasis were terminated due to adverse events, mainly affecting the gastrointestinal tract, which were caused, at least in part, by poor selectivity versus the different PDE4 subfamilies. Because of the high similarity between the isoforms of PDE4, there is a lack of isoform selective inhibitors, which may improve tolerability. By contrast, however, since PDEs are also expressed in structural and inflammatory cells in the respiratory tract and may contribute to disease development and exacerbation [16], a selective targeting of PDE4 may not be fully effective. In such conditions, the simultaneous inhibition of multiple PDE families would be a promising direction for the discovery of novel PDE4 inhibitors. Finally, for treating respiratory diseases, inhaled PDE4 inhibitors may reduce unwanted side effects by directly targeting the airways while limiting systemic exposure, as exemplified by clinical studies with Tanimilast [11].

For an extensive review on PDE4 inhibitors development, usage, role and side effects in therapy, which goes beyond the scope of this review, we refer the reader to the recent and comprehensive literature (e.g., [4,7,14,15,17,18,19,20,21]).

## 3. DC Subtypes and Functions in the Respiratory Tract

### 3.1. The Biology of DCs

#### 3.1.1. DC Subpopulations

Dendritic cells are a heterogeneous group of cells originating in the bone marrow from both myeloid and lymphoid precursors. DCs are identified by the abundant expression of major histocompatibility complex class II (MHC-II) molecules and the absence of lymphocytic, monocytic and granulocytic-specific makers [22]. According to their ontogeny, phenotypical features, tissue distribution and transcriptional profiles, DCs are classified into different subsets: conventional or classical DCs (cDCs), including cDC1s and cDC2s; plasmacytoid DCs (pDCs); and monocyte-derived DCs (moDCs) [23,24,25]. cDC1s require the transcription factors IRF8, ID2 and BATF3 for development and can be distinguished by the expression of the membrane markers CD141/Thrombomodulin/BDCA-3 and Clec9A [26]. The transcription factor IRF4 is essential for cDC2s that can be phenotypically distinguished by the expression of the membrane markers BDCA1/CD1c and SIRPa [26]. Single-cell analysis revealed an additional level of complexity in DC heterogeneity by identifying multiple cDC2 subsets, such as DC2 and DC3 [27,28], whose developmental origin and functional properties need further investigations [29,30]. pDC differentiation relays on the transcription factors TCF4, IRF8 and Ikaros family zinc finger 1 (IKZF1) [31,32,33] and can be identified by the phenotypic markers BDCA-2/CD303, BDCA-4/CD304, ILT7 and the receptor for IL-7 [34]. Finally, moDCs represent a DC subset originating from blood monocytes recruited in the tissues in response to inflammatory stimuli [35].

#### 3.1.2. DC Maturation

DCs commonly reside in blood, non-lymphoid and lymphoid tissues with the unique role among immune cells to activate and tailor T cell responses to combat specific pathogens and foreign antigens [36]. From a functional point of view, DC can be classified in immature and mature cells (Figure 1).

Immature cDCs reside in the peripheral tissues and express chemokine receptors, such as CCR1, CCR5 and CCR6, allowing them to be recruited by inflammatory chemokines produced by injured cells. Immature cDCs patrol the local environment through the expression of a vast repertoire of innate receptors, which recognize different types of exogenous or endogenous danger signals [24]. In response to this recognition, DCs undergo a profound and ordered alteration of their functional status and tissue localization, known as maturation. During maturation, cDCs lose their endocytic and phagocytic abilities, rearrange chemokine receptor repertoire and re-program the expression of cytokines and costimulatory and MHC molecules to specifically prime T cells [37,38]. On these bases, maturation markers include antigen-presentation molecules (MHC-I and II), costimulatory molecules (such as CD80, CD83 and CD86) and homing lymph node chemokine receptors (CCR7 and CXCR4) [39]. Indeed, maturing DCs leave peripheral tissues and migrate toward regional lymph nodes (Figure 1). Here, cDC1s specialize in the processing and presentation of intracellular antigens and in shaping anti-viral immune responses by cross-presenting virus-associated antigens to CD8+ T cells via MHC-I [40]. In parallel, cDC2s efficiently present MHC-II-associated antigens to CD4+ T cells, promoting Th1, Th2 and Th17 polarization [41]. By contrast, pDCs mature in response to viral or tumor stimuli accumulate in non-lymphoid tissues and secrete huge amounts of type I interferons (IFNs). Mature pDCs can also prime T cells but with a lower efficiency in respect to cDCs [42].

Notably, DC maturation is no longer considered a one-step event. Indeed, the endogenous production of secondary mediators by activated DCs acts as an essential amplification loop besides stimulation by pathogen recognition [43]. Among these, type I IFNs play a key role in both cDC and pDC maturation [44,45], while TNF-α and IL1β are essentials for cDCs [46,47].

#### 3.1.3. Tolerogenic DCs

DCs, and especially cDC1s [48], are also crucial for the establishment of peripheral tolerance in steady-state conditions [48]. Tolerogenic DCs express low levels of co-stimulatory and pro-inflammatory molecules and may promote T cell anergy or clonal deletion, as well as the conversion into regulatory T cells (Tregs) [48,49]. Interestingly, at specific barrier sites, such as the airways, extrinsic signals can also induce the tolerogenic differentiation of DCs to maintain immune homeostasis in the presence of commensal organisms and other antigens, even under partially pro-immunogenic conditions [48,50]. pDCs may also contribute to peripheral tolerance [51], in particular when antigen uptake and presentation occurs in conditions that also trigger pDC inhibitory receptors such as BDCA2 and DCIR [52,53].

Tolerogenic DC-based therapies are being actively investigated for the management of aberrant inflammatory or immune-mediated conditions [54]. As of today, tolerogenic DCs can be induced by treatment with immunomodulatory agents, such as corticosteroids or calcineurin inhibitors, which impair phenotypical and functional maturation and induce regulatory markers (e.g., PD-L1, ILT3, etc.) and mediators (e.g., IL-10 and TGF-β), as well as indoleamine 2,3-dioxygenase 1 (IDO1), which suppresses T cell activation by promoting tryptophan metabolism [50]. Here, we collected evidence that PDE4 inhibitors induce a novel type of tolerogenic DCs that are clearly distinct from those induced by corticosteroids (see Section 4).

### 3.2. DCs in the Respiratory Tract

The frequency and the distribution of the different DC subsets is a function of the tissue site [55] and, especially at barrier sites, is finely tuned to maintain the balance between protective immunity and self-tolerance [37,56].

cDCs are present in normal human airway mucosal epithelium, lung parenchyma, interalveolar septa and visceral pleura [57] (Figure 2). In the lung parenchyma, the frequency of cDCs is particularly high with respect to other mucosal sites, with cCD2s surpassing cDC1s. cDC2s and pDCs are particularly enriched in lung lymph nodes as compared to other lymphoid tissues [55]. Concerning the airways, DCs lie along the mucosal epithelium, spanning from the upper to lower respiratory tract and forming an extensive network to continuously sample antigens from the environment [56]. DC density decreases from large, conducting airways to respiratory bronchioles, where cDC1s are mostly associated with respiratory epithelium, sampling airway content by protruding their dendrites through epithelial tight-junctions (Figure 2). Interestingly, studies in mice and in a 3D model of the human airway showed that DC build tight-junction-like complexes with epithelial cells in order to preserve epithelial integrity [58,59]. By contrast, cDC2s are located in the basal layer of the epithelium. DCs in the visceral pleura were also described to arrange in parallel with the pleural surface, with extensive cytoplasmic processes projected to the mesothelial lining [57].

Under homeostatic conditions, both cDC1s and cDC2s in the airways exhibit an immature phenotype. Upon inflammation, mature cDC2s, and with a lower frequency cDC1s, migrate to the local lymph node, where cDC1s localize preferentially in the T cell zone while cDC2s tend to aggregate in proximity to B cell follicles [55]. Additionally, circulating- and monocytic-DC precursors are rapidly recruited to the site of infection via chemotactic signals [60]. In some cases, the influx of DC precursors occurs within two hours after challenging, being even faster than the infiltration of neutrophils [61]. Together with pulmonary DCs, these cells capture, process and transport antigens to the draining pulmonary lymph nodes where they activate and imprint the magnitude of the immune response [56].

Given the central role of DCs in immune regulation, as well as their strategic anatomical location, it is not surprising that the dysregulation of DC functions may represent a pathogenetic determinant, as well as a relevant pharmacological target in different lung diseases [62], as described in Section 5.

## 4. DC Regulation by PDE4 Inhibitors

The effects of PDE4 inhibitors on maturation and immune functions of DCs has been investigated in a number of studies revealing the modulation rather than the suppression of the proinflammatory and T-activating capabilities of DCs. These changes are detailed in the following paragraphs and depicted in Figure 3.

### 4.1. DC Phenotype

DC activation implies the upregulation of several signature cell surface molecules, marking their transition into the so-called “mature” state. Different from other anti-inflammatory or immuno-modulatory agents, such as corticosteroids, different PDE4 inhibitors, including Tanimilast and Apremilast, neither fully abrogated the acquisition of a mature phenotype nor promoted a classical tolerogenic profile in inflamed DCs [6,63,64,65,66] (Figure 3, upper panels). Conversely, PDE4 inhibitors selectively restrained the upregulation of MHC-I and CD80 while sparing the increase of MHC-II, CD83 and CD86. The lymph node migratory capacity was also retained as the result of the untouched upregulation of chemokine receptors CCR7 and CXCR4. Of note, similar results were obtained when DCs were treated with cAMP analogs [67]. Unexpectedly, DC inhibitory molecules such as PD-L1 and ILT3/CD85k were downregulated by Tanimilast. These inhibitory molecules are induced during inflammatory DC activation, possibly as a feedback loop to prevent excessive T cell responses, at least partially as a result of the autologous secretion of inflammatory mediators. Thus, reduction of proinflammatory mediators by PDE4 inhibitors (see Section 4.2.1.) may block this feedback circuit, resulting in PD-L1 and ILT3/CD85k reduction [65,68,69]. The biological significance of this phenomenon remains to be explored.

Of particular interest, we found that Tanimilast, but not the glucocorticoid budesonide, could strongly induce the expression of CD141 both in moDCs and in circulating cDC2s [65]. CD141 is a transmembrane protein that is abundantly expressed by endothelial cells, where it is regulated by the levels of intracellular cAMPs [70,71]. In this cell type, CD141 exerts various and well-characterized anti-coagulant and anti-inflammatory activities, such as the sequestration of LPS and/or HMGB1 and degradation/inhibition of complement deposition [72,73]. CD141 is also a marker of cDC1 cells, a subpopulation of human DCs known for their superior antigen cross-presenting capacity [74]. However, the function of CD141 in DCs remains unknown. Recently, the lectin domain of this molecule was suggested to play a role in DC-mediated immune stimulation in the lung [75]. Consistent with this view, CD141^+^ DCs within peripheral tissues were shown to display a tolerant phenotype [76,77] to preferentially induce type 2 immunomodulatory responses [78] and even potent regulatory T cells [76]. Despite the fact that we could not identify the role of CD141 upregulation in DC activated in the presence of Tanimilast, we found that the percentage of CD141^+^ correlated with the uptake of dead cells, a function that is crucial to trigger antigen cross-presentation. In addition, we have proposed that CD141 may represent a marker of Tanimilast-induced immunomodulatory DCs [65].

### 4.2. DC Immune Functions

#### 4.2.1. Production of Pro-Inflammatory and T-Polarizing Mediators

Mature DCs secrete a wide range of pro-inflammatory cytokines and chemokines, which, in turn, initiate and amplify inflammation via the recruitment of innate inflammatory cells and the polarization of T lymphocytes [79]. In this context, the administration of PDE4 inhibitors during DC maturation potently prevented the secretion of prototypic pro-inflammatory cytokines (TNF-α, IL-1β and IL-6) and chemokines that are essential for the influx of immune cells into inflamed tissues (CCL3, CXCL9 and 10) [6,63,64,65,66] (Figure 3, middle panels). PDE4 inhibitors, including Tanimilast, effectively suppressed Th1/Th17-polarizing cytokines (IL-12, type I IFNs, IL-6 and IL-23) and strongly enhanced the production of CCL22, a Th2/Treg attractor [63,64,65]. Tanimilast also increased the expression of cAMP-dependent immunosuppressive molecules, such as IDO1, TSP-1, VEGF-A and Amphiregulin, in stimulated DCs [65], which decrease T cell activation, skew towards a Th2/Treg profile and promote the resolution of inflammation [80,81,82,83,84,85].

#### 4.2.2. T Cell Activation

One prominent characteristic of DCs is their instructive role in the activation of adaptive immunity, depending on their unique capability to activate and polarize naïve T cells [39]. To elucidate the effects of PDE4 inhibition on this DC function, several studies exploited allogeneic mixed lymphocyte reaction experiments, which involve the coculture of DCs matured in the presence or absence of PDE4 inhibitors with allogeneic T lymphocytes [6,63,64,65,66] (Figure 3, lower panels).

DCs matured in the presence of PDE4 inhibitors displayed a profound impairment in Th1 induction but, at difference with budesonide, still retained their ability to grossly expand naïve CD4+ T cells [6,64,65]. This is in line with the above-described blockade of IL-12 together with the untouched retention of MHC-II, CD83 and CD86 upregulation. Moreover, Tanimilast increased the percentage of Th2 cells, which positively correlated with the CD86/CD80 ratio [64,65], as previously described in different experimental settings [86,87]. Indeed, the Th2 biased effects were not only observed in human DCs treated with PDE4 inhibitors but also in those treated with other cAMP-elevating agents, such as cholera toxin, prostaglandins and others [88,89,90,91]. However, it is important to stress that the suppressive effects of PDE4 inhibitors on the Th-1/Th-17 axis were mostly demonstrated in the context of type 1 inflammation. Further studies are therefore necessary to fully explore the modulatory effects of these agents on Th2/Th17-mediated conditions.

In marked contrast with the human setting, Roflumilast and cAMP-elevating agents were shown to favor the Th17-polarizing properties in murine bone marrow-derived DCs [92,93] by efficiently inhibiting the production of IL-12 and promoting a higher level of Th17-expanding cytokines IL-23 and IL-10. Interestingly, Kim and colleagues proposed a novel signaling network in which low levels of cAMP in murine DCs promoted Th2 immunity, whereas increased cAMP concentration induced Th17 response [94]. Therefore, the effect of PDE4 inhibition on the polarizing potential of DCs may differ among DC subsets and species, and results may need to be interpreted carefully depending on the different experimental models and settings.

At difference with CD4^+^ T cells, CD8^+^ T cell proliferation and effector functions were strongly inhibited by Tanimilast at a level similar to budesonide accordingly to the reduced expression of both IL-12 and MHC-I molecules [64,65]. These data were indeed in line with other reports on cAMP-elevating agents, including NECA, forskolin and IBMX, which were shown to limit the expansion of Melan-_A26–35_ specific CD8+ T cells by modulating the process of DC activation [95].

Altogether, the existing literature indicates that PDE4 inhibitors confer to DCs a distinct phenotype, highlighted by the upregulation of CD141 and associated with tolerogenic functions distinct from those of classical immunosuppressive molecules such as corticosteroids (Figure 3). Type 2 immunity is normally linked to host defense against helminths and allergy. Currently, however, type 2 responses are being recognized as playing a broader role in immune surveillance at tissue barrier sites, repair responses and the restoration of homeostasis [96]. In particular, a Th2-skewed T cell activation along with the expression of resolution-inducing molecules may define DCs conditioned by PDE4 inhibitors as pro-resolving mediators, potentially relevant in the limitation of dysregulated immune responses as well as the restoration of homeostasis at sites of injury [97]. Thus, cAMP signaling in human DCs may play a pivotal role in the differential steering of CD4^+^ T cell activation as well as in enabling the signals for the resolution phase of inflammation. These results support the use of PDE4 inhibitors as a disease-modifying strategy to manage certain inflammatory or immune-mediated respiratory diseases also via the increase of intracellular cAMP in DCs [98].

## 5. Pathological Role of DCs in Respiratory Diseases and Implications for PDE4 Inhibitor Therapy

### 5.1. COPD

COPD is a chronic lung inflammatory condition characterized by persistent airway obstruction and remodeling, with smoking being the most important risk factor of the disease [99]. Several studies demonstrated a link between DCs and the pathogenesis of COPD. The accumulation of DCs in the airways with activation of Th1/Th17 lymphocyte subsets and overactivation of CD8+ T cells were consistently observed during disease progression and aggravation [100,101]. In particular, the increased cytotoxic activity of CD8+ T cells was associated with an increased expression of Clec9A, consistent with the function of cDC1s [101]. Moreover, cDC2s purified from a COPD patient displayed an increased ability to induce follicular T helper cells, suggesting their contribution to the formation of tertiary lymphoid tissues, which play a detrimental role in chronic inflammatory conditions, including lung diseases [102].

As of now, Roflumilast is the only PDE4 inhibitor approved by the FDA for the treatment of COPD, while Tanimilast is advancing in the phase III clinical trials with promising results for its potency and safety profile [103]. Indeed, PDE4 inhibitors displayed promising anti-inflammatory effects in experimental models for COPD in vitro and in vivo [11,104].

In particular, the administration of PDE4 inhibitors efficiently reduced the influx of neutrophils to murine lungs acutely exposed to cigarette smoke and restrained the progression to emphysema in mice with chronic smoke exposure [105,106]. A pivotal study with cilomilast revealed the decrease of CD8^+^ T cells and CD68^+^ macrophages, two critical components associated with COPD pathogenesis, in the bronchial specimens of COPD patients [107]. In a later study, Roflumilast induced a substantial decrease of soluble inflammatory mediators in sputum, thus partly contributing to the amelioration of lung functions [108]. Finally, in accordance with in vitro results described in previous section, a randomized study revealed an enhanced expression of genes related to immunosuppression and Th2 response, including TSP-1, VEGF-A, Amphiregulin and CD141 in sputum cells of COPD patients treated with Tanimilast on top of inhaled corticosteroids [64,103] (Figure 3, upper panels).

Despite direct experimental evidence still lacking, it is tempting to hypothesize that, by modulating DC activation, PDE4 inhibitors may create a local microenvironment less supportive to Th1/Th17 and more favorable to Th2 differentiation, which was demonstrated to enhance the therapeutic response to inhaled corticosteroids [109]. These observations provide a mechanistic rationale for the administration of PDE4 inhibitors as add-on therapy in a defined subgroup of COPD patients who remain symptomatic upon standard-of-care treatment. This approach was also investigated in randomized trials with Roflumilast, where patients on combination therapy displayed improved outcomes compared to monotherapy [110].

### 5.2. Asthma

Asthma is a chronic inflammatory condition that causes reversible airway obstruction, resulting in occasional breathing difficulties and wheezing in response to an array of environmental triggers. In terms of pathogenesis, a dominant Th2 paradigm, together with eosinophilic inflammation in the airway, are recognized as the main underlying molecular mechanisms of the disease [111]. However, a Th2-low endotype driven by distinct, poorly characterized molecular mechanism and featuring neutrophilic or paucigranulocytic inflammation is also described in various asthma subtypes. Interestingly, this clinical presentation is responsible for the majority of cases resistant to inhaled corticosteroids and short-acting β2 agonist [112].

The role of DCs was investigated, especially in the Th2 endotype of asthma. In allergic asthma, both cDC1s and cDC2s were found to be increased in induced sputum after allergen inhalation [113,114]. A single-cell proteomic and transcriptomic profiling of nasal biopsies demonstrated a different behavior of cDC2 from allergic vs. non-allergic patients, characterized, respectively, by inflammatory vs. inhibitory/tolerogenic transcriptional changes upon allergen challenge [115]. Mice lacking lung DCs failed to initiate and to perpetuate allergen-driven airway inflammation [116,117,118]. A single-cell RNA-sequencing study in a mouse model of allergic asthma reported an imbalance of two newly described cDC1 clusters toward a pro-inflammatory versus a tolerogenic phenotype [119]. Interestingly, the adoptive transfer of tolerogenic DCs in models of allergy-induced persistent suppression of Th2 allergic response, thus conferring long-term protection from airway inflammation [120,121,122]. Likewise, restraining DC maturation not only alleviated lung injury but also accelerated the recovery phase of the disease [123,124,125]. Despite the role of pDCs being scarcely investigated, mouse data indicated that pDCs are able to prevent asthmatic reactions to harmless inhaled antigens [126].

Preclinical evidence of PDE4 targeting in a rat model of allergic asthma indicated that ovalbumin sensitization and challenge significantly increased the activity of PDE4 and mRNA expression of different PDE4 isoforms, including PDE4A, PDE4C and PDE4D [127]. Recently, PDE4B was shown as the main player in driving Th2 development in allergic asthma: its knockdown reduced the number of Th2 cells and cytokines (IL-4, IL-5 and IL-13) and switched off airway hyperresponsiveness upon allergen challenge [128]. Selective blocking of PDE4 by rolipram in ovalbumin-sensitized animal models not only counteracted the increase of eosinophils in the blood and BAL but also relieved the symptoms of airway hypersensitivity [129]. Especially, increasing cAMP levels in dendritic cells of an allergic model restrained Th2 development and airway inflammation in both in vitro and in vivo experiments [130]. In humans, asthmatic patients under PDE4 inhibition therapy displayed attenuated allergen-induced airway inflammation, along with lower accumulation of inflammatory cells and cytokines [131].

Whether PDE4 inhibitors exert their therapeutic effects on Th2-asthma by regulating the immunogenic features of DCs remains to be clearly defined, although one study showed that the modulation of cAMP levels in DCs reduced Th2 development and airway inflammation in a bronchial asthma model [130] (Table 2). The data presented above indicate that human DCs, when treated with PDE4 inhibitors, acquire a semi-mature phenotype endowed with a Th2-skewing capacity, which seems inconsistent with such therapeutic exploitation. However, in all in vitro studies, DCs were activated with LPS, which induces Th1 responses, thus failing to recapitulate the model of DC activation occurring in allergic asthma. Further studies are required in which DCs are activated with allergens or TSLP, a molecule that is emerging as a pivotal inducer of different types of asthma [132]. In addition, in vitro studies on DCs cannot provide reason for the direct effects of PDE4 inhibition on other target cell types that cooperate in the establishment of the immunomodulatory effects of PDE4 inhibition observed in in vivo models [6,63,64,65,66]. Indeed, in vitro proliferation experiments showed that PDE4 inhibitors also directly impact T cell activation [133,134,135]. On the other hand, a dual anti-inflammatory/Th2-orienting effect could be useful in the treatment of pathologies characterized by excessive Th1/Th17 responses such as Th2-low asthma. This speculation is supported by the lower number of neutrophils and decreased airway inflammation reported in neutrophilic asthmatic models under Roflumilast therapy [136]. The efficacy was largely enhanced when steroids were added to PDE4 inhibitors, proposing a new strategy of drug combination in asthma management.

Another interesting immunomodulatory feature of PDE4 inhibitors on DCs is their capacity to induce high levels of CD141, since CD141^+^DCs were suggested to exert protective effects against bronchial asthma [65,75,137]. In in vivo asthma models, the adoptive transfer of CD141^+^ DCs or administration of inhaled soluble CD141 resulted in a decline of inflammatory cells, eosinophils and Th2-related cytokines, thus reducing disease severity and progression [75]. Even though a number of studies reported an elevation of CD141 expression in peripheral blood associated with air limitation in asthmatic patients, it appeared to be a compensatory response more than a pathogenic mechanism [75]. Despite it being tempting to speculate that CD141^+^ DCs induced by PDE4 inhibitors may present beneficial effectors in the management of asthma and other allergic diseases, further studies are required to confirm this hypothesis.

### 5.3. Coronavirus Disease 2019 (COVID-19)

COVID-19 is an infectious disease caused by severe acute respiratory syndrome coronavirus 2 (SARS-CoV-2) [138]. As the lung is one of the preferential targets of the virus, most patients show bilateral pneumonia on admission to healthcare facilities, and one fifth of them progress to severe lung injury [139]. It is now clear that the progression and the severity of COVID-19 is a direct consequence of dysregulated host immune responses rather than of viral replication per se. These responses include hyper-inflammation and cytokine storm, leading to uncontrolled diffuse tissue damage and acute organ failures [140,141,142,143]. Several prototypic immunomodulatory agents have been tested in clinical trials, and yet none were proven to be fully effective for the management of COVID-19, especially at later stages of the disease [144,145].

Several groups also proposed PDE4 inhibitors for COVID-19 management [3,146,147] based on the assumption that the upstream inhibition of inflammatory pathways via increased cAMP levels may outclass single anti-cytokine drugs, especially once multiple cytokines are massively released [148]. In vitro, our group showed that human DCs activated by SARS-CoV-2 RNA in the presence of Tanimilast massively decreased multiple pro-inflammatory chemokine and cytokine production and induced defective development of Th1/Th17 and CD8+ effector cells while increasing Th2 cells [64]. Of note, Th2 cytokines were shown to decrease the expression of the SARS-CoV-2 receptor angiotensin-converting enzyme 2 [149,150,151] and to sustain the anti-thrombotic properties of airway endothelial cells [152]. Interestingly, SARS-CoV-2-induced coagulopathy may benefit from the anti-coagulant activity of CD141^+^ DCs, which were shown to directly activate two critical components of anti-coagulation pathways, protein C and proCBP2 [153]. Thus, the immuno-modulatory effects of PDE4 inhibitors on DC activation may help counteract key pathological mechanisms in severe SARS-CoV-2 pneumonia, such as the overactivation of innate cells and T lymphocytes and hypercoagulation [152,154,155,156,157].

In accordance, PDE4 inhibitors, such as Apremilast, when administered actively for the treatment of COVID-19 or passively as a targeted therapy of underlying diseases, displayed rapid and positive effects in a subgroup of severe SARS-CoV-2 pneumonia [158,159,160]. However, the therapeutic efficacy of PDE4 inhibitors in the clinical practice requires broader investigation and analysis. To this aim, a phase III clinical trial with Apremilast has been completed (https://classic.clinicaltrials.gov/ct2/show/NCT04590586, accessed on 1 July 2023), but results are not yet available to confirm its applicability in the management of COVID-19.

### 5.4. Acute Lung Injury (ALI) and Acute Respiratory Distress Syndrome (ARDS)

Although ALI and ARDS are caused by different injuries and conditions, they are studied as a single entity because of their similar clinical picture, consisting of acute systemic inflammation, bilateral pulmonary infiltrates and severe hypoxemia requiring critical care and ventilation support [161]. In LPS-induced ARDS models, pulmonary DCs facilitated Th1 response and enhanced the infiltration of neutrophils, which are essential in the initiation and amplification of severe lung injury [162,163,164].

Numerous in vivo studies investigated the preventive and therapeutic potential of PDE4 inhibition for these pathological conditions. For example, lung edema and pulmonary functions of rabbits induced to ALI by saline administration were significantly improved by Roflumilast [165], and other PDE4 inhibitors performed similarly in models where ALI was induced by chemicals or microbial agents [166,167,168]. In mice infected with a lethal dose of Influenza A H1N1 virus, rolipram, in combination with antiviral therapy, significantly reduced lung injuries and increased the survival rates up to 80–100% [169]. Importantly, the adoptive transfer of Resveratrol-treated DCs to WT mice before LPS challenge enhanced survival rates, reduced lung tissue damage and lowered the expression of Th17 cells in the lung [123] (Table 2).

In line with in vivo research, in vitro data focusing on human DCs provided supportive evidence for the application of PDE4 inhibitors in ALI/ARDS because of their capacity to restrain DC maturation [123,124,125], Th1/Th17 polarization and secretion of crucial inflammatory mediators [6,63,64,65,66]. For example, Resveratrol, exerting its anti-inflammatory effects through PDE inhibition [170], alleviated LPS-induced lung injury by regulating DC immunogenic properties [123]. In addition, because the inflammatory milieu is enriched in HMGB1 and Thrombin, the induction of CD141 may contribute in scavenging these danger signals [11,104].

Despite promising results, the use of PDE4 inhibitors is not included in the current treatment guidelines of ALI/ARDS because of insufficient clinical data and a narrow therapeutic window. In this scenario, inhaled PDE4 inhibitors may represent a valuable alternative to increase tolerability and efficacy.

## 6. Conclusions

DCs represent a paramount checkpoint of the immune system, being immunogenic to activate adaptive immunity upon pathogen encounters and also serving as an active inducer of tolerance in peripheral tissues at a steady state. The modulation of DC properties is an innovative modality for treating inflammatory and immune-mediated conditions because it targets the underlying disease mechanism. In this review, we highlighted PDE4 inhibitors as promising DC-modifying agents and the potential fallout of this modulation in a number of respiratory diseases where DCs play critical pathogenic roles. Despite a few studies directly addressing DC modulation as the relevant mechanisms of action substantiating the effects of these drugs, we suggest that DCs may deserve more attention in the future as targets of PDE4 inhibitors in multiple respiratory conditions associated with chronic and acute airway inflammatory responses.

## Figures and Tables

**Figure 1 pharmaceutics-15-02254-f001:**
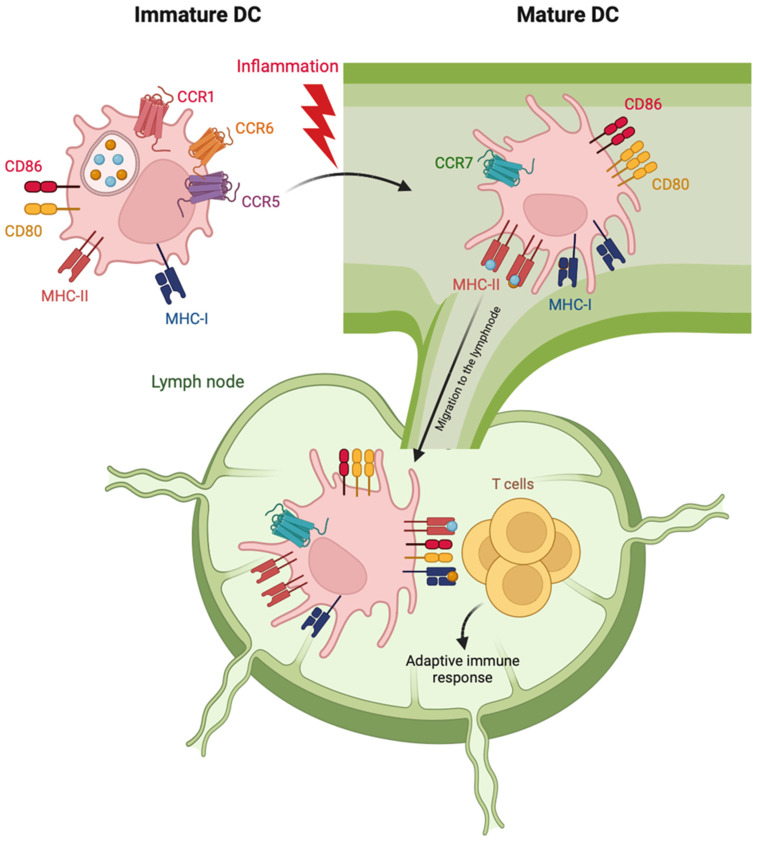
Schematic representation of phenotypic features of cDC maturation. Following activation stimuli in the peripheral tissues, maturing cDC starts moving toward lymphatics by down-regulating the expression of receptors for inflammatory chemokines (CCR1, 5 and 6) and upregulating the lymph node-addressing chemokine receptor CCR7. The increased expression of MHC molecules allows mature cDC to present engulfed antigens to T cells. The increased expression of costimulatory molecules (CD80 and 86) on mature cDC promotes the activation of antigen-specific T cells and the adaptive immune response.

**Figure 2 pharmaceutics-15-02254-f002:**
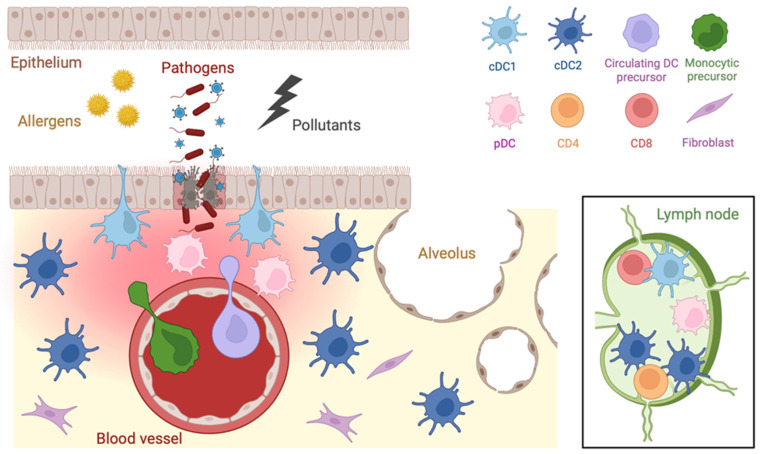
Distribution of DCs in the respiratory tract. Tissue resident DCs are differentially located along the airways and in the regional lymph nodes. Upon inflammation, DC precursors originating from blood monocyte or circulating DC and plasmacytoid DC are rapidly recruited into the damaged tissue.

**Figure 3 pharmaceutics-15-02254-f003:**
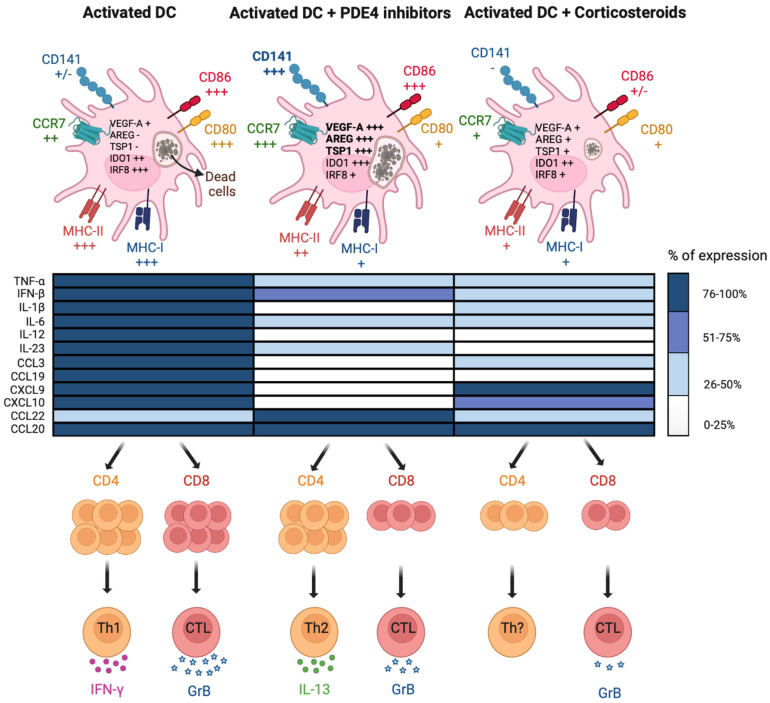
Distinctive immunomodulatory functions of PDE4 inhibitors on cDCs. PDE4 inhibitors and corticosteroids differently affect cDC maturation induced by TLR ligands and their ability to activate T cells. Upper panels depict the modulation of membrane markers, functional features (dead cell uptake) and gene expression in activated cDCs. Bold characters indicate the parameters that have been found modulated in the same way in sputum cells of COPD patients treated with Tanimilast on top of standard-of-care therapy. Middle panel summarizes the modulation of cytokine and chemokine secretion in activated cDCs. Data in the heat map are expressed as the % of the maximal stimulation over control. Lower panels illustrate the modulation of T cell proliferation/effector functions. Most of the results shown in the pictures were obtained with Tanimilast and budesonide or dexamethasone.

**Table 1 pharmaceutics-15-02254-t001:** PDE inhibitors in ongoing clinical development for respiratory diseases.

Drug Name	Company	Indications	Phase	NCT Number	Comments
Roflumilast (Daliresp)	AstraZeneca	COPD	n/a	Approved	Approved in EU (2010) and in the USA (2011) for the management of severe COPD.
		Bronchiec- tasis	II	NCT03988816	Efficacy on lung function and mucus properties.
Tanimilast (inhaled)	Chiesi Farmaceutici	COPD	III	NCT04636801 (Pilaster)	Placebo-controlled (Pilaster) and active (Roflumilast)-controlled (Pillar) 52-week studies for efficacy and safety as an add-on to maintenance triple therapy in COPD and chronic bronchitis.
			III	NCT04636814 (Pillar)	
		Asthma	II	NCT01689571	Efficacy, tolerability and safety of Tanimilast in asthmatic patients.
Ensifentrine (RPL554)	Verona Pharma	COPD	II	NCT05270525	Under evaluation as an add-on to standard of care treatments.
		Asthma	II	NCT02427165	
		COVID-19	II	NCT04527471	
BI 1015550	Boehringer Ingelheim International	Idiopathic Pulmonary Fibrosis	III	NCT05321069	Evaluate long-term efficacy and safety of BI 1015550.
Apremilast (Otezla)	Amgen	COVID-19	III	NCT04590586	Efficacy and safety of Apremilast as add-on to standard of care in hospitalized patients.
Cilomilast (Ariflo)	GSK	COPD	III	NCT00103922	Completed 24-week study for safety and efficacy in COPD patients (development terminated).
GSK256066 (inhaled)	GSK	COPD	II	NCT00549679	Safety and tolerability in mild to moderate COPD patients (development terminated).
Tetomilast (OPC-6535)	Otsuka Pharmaceutical	COPD	II	NCT00917150	24-month study for efficacy and safety of OPC-6535 in COPD patients (last update 2021).
Oglemilast	Forest Laboratories	COPD	II	NCT00671073	14-week study for safety and efficacy of a range of Oglemilast doses (last update 2019).
		Asthma	II	NCT00322686	Prevention of bronchoconstriction after the administration of allergen in mild asthma patients (last update 2012).
Revamilast	Glenmark Pharmaceuticals Ltd. India	Asthma	II	NCT01436890	12-week study for effects of Revamilast in patients with chronic persistent asthma (last update 2013).

**Table 2 pharmaceutics-15-02254-t002:** Therapeutic effects via modulation of cAMP levels in DCs.

Animal Model	Findings	Reference
Bronchial asthma model (Gnas^ΔCD11c^ mice)	Adoptive transfer of OVA-loaded DCs from Gnas^ΔCD11c^ mice induced Th2 response and airway inflammation in WT and Gnas^ΔCD11c^ mice. Adoptive transfer of 8-CPT-cAMP-treated DCs from Gnas^ΔCD11c^ mice reduced Th2 development and airway inflammation in recipient mice.	[130]
LPS induced ALI in mice	Adoptive transfer of Resveratrol-treated DCs to WT mice before LPS challenge enhanced survival rates, reduced lung tissue damage and lowered the expression of Th17 cells in the lung.	[123]

## Data Availability

Not applicable.

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
