# Peer review of "Modulation of Human Dendritic Cell Functions by Phosphodiesterase-4 Inhibitors: Potential Relevance for the Treatment of Respiratory Diseases"

_pharmaceutics, 2023, doi:10.3390/pharmaceutics15092254_

Round 1
Reviewer 1 Report
This article titled as “Modulation of Human Dendritic Cell Functions by Phospho-diesterase-4 Inhibitors: Relevance to Respiratory Diseases” discussed the evidence that phosfodiesterase-4 inhibitor (PDE4i) modulate inflammatory DC activation by decreasing the secretion of inflammatory and Th1/Th17 polarizing cytokines, although preserving the expression of costimulatory molecules and the CD4+ T cell-activating potential. Authors also pointed out that DCs activated in the presence of PDE4is induce a preferential Th2 skewing of effector T cells, retain the secretion of Th2-attracting chemokines and increase the production of T cell regulatory mediators such as IDO1, TSP-1, VEGF-A and Amphiregulin. However, the evidence summarized by Authors seems too week to support their hypothesis, especially the actual effects of PDE4i in treating selected diseases via modulating DC functions. Moreover, the advantages of applying PDE4i in treating respiratory diseases compared to conventional therapy have not been clearly displayed and discussed, making the very limited novelty of this article. Other apparent flaws have been listed as following:
1.The existed PDE4is and those who are currently in the going clinical trials for treating selected diseases (COPD, asthma, covid 19, ARDS), should be summarized in a table. Information such as clinical outcome, ORR (Objective response rate) should all be displayed.
2.In the end of this manuscript, author eventually claimed that “PDE4is as promising DC-modifying agents and their therapeutic potential in several respiratory diseases where DCs play critical pathogenic roles, suggesting that DC modulation may represent one relevant mechanisms of action substantiating the effects of these drugs in multiple respiratory conditions associated with chronic and acute airway inflammatory responses.”
a. Authors have concluded some therapeutic effects of PDE4is on selected diseases. Please also displayed actual evidence that proves the therapeutic effects via PDE4is modulating DC. Relevant information should also be summarized in a table.
b.Please also discuss the potential disadvantages of using PDE4is and solutions.
c.Please systematically describe the potential roles of PDE4is in Respiratory Diseases treatment.
3.Authors should create a single section to specifically discuss the advantages of using PDE4i in treating respiratory diseases compared to conventional therapeutic strategies.
4. Please also evaluate the necessity of developing PDE4i in treating respiratory diseases.

Reviewer 2 Report
One can agree with the authors of the article regarding the predominant immunosuppressive and anti-inflammatory effects of cAMP and, consequently, phospho-2-diesterase-4 inhibitors on DCs.
However, a certain degree of internal inconsistency of these effects must be taken into account. For example, many of these cAMP effects are associated with the phosphorylation of specific PKA substrates, but this is a very complex pathway. There are at least nine membrane-associated ACs and one cytosolic Ca2+-sensitive AC, eight families of phosphodiesterases that hydrolyse cAMP, as well as three catalytic and four regulatory PKA subunits, and up to six cAMP-dependent ion channels. In addition, all nine isoforms of membrane ACs can be modulated (activated or inhibited) by Ca2+, either directly or indirectly via Ca2+-binding proteins such as calmodulin (CaM), CaM kinases (CaMK), calcineurin (CaN), PKC and Ras small GTPases. All this makes the effects of cAMP on individual cell types very diverse, complex and often internally contradictory. It should be borne in mind that cAMP and calcium mechanisms are rarely independent, but can often be antagonistic, but sometimes synergistic or overlapping. For example, one of the characteristics of cAMP is its ability to inhibit proliferation in many cell types, but stimulate proliferation in others. At the same time, cAMP has a specific effect on cell type, influencing their proliferation and differentiation largely through the cross-talking but ambiguous interaction of cAMP with the Ras/Raf/MEK/ERK signalling pathway, including in DCs. In some cells, PKA can modulate not only MAPK-ERK, but also phosphorylate and activate p38 MAPK, phosphorylate and activate PKC. In addition, by activating phosphorylation of CaMKII and ERK, PKA can stimulate many cellular stress signalling pathways in cooperation with mobilised calcium, PKC and Src kinase. These data do not allow us to clearly identify the cAMP/PKA pathway as an inhibitor of proinflammatory cellular stress and immune reactivity, but rather to talk about the complex modulatory function of cAMP/PKA on various proinflammatory and immunotropic mechanisms. At the same time, we can only talk about the predominant action of cAMP inducers in relation to immune processes. However, the direction of this action needs to be clarified in each specific situation.
Obviously, all of the above applies to DCs. Although PKA signalling significantly suppresses cytokine production, the cumulative effect of PKA activation leads to increased DC activation of allogeneic T cells [PMC3100203]. PGE2 affects the production of IL-23 in human DC and this effect is dependent on the cAMP signalling pathway [PMC4564649]. cAMP signalling pathways can enhance or diminish c TCR signalling [PMC6684054]. The cAMP-PKA-CREB pathway in cDC2 stimulates Th17 generation and inhibits Th2 induction [PMC7000221]. Of course, there are many other references in the scientific literature that show the ambiguity of the effect of increased cAMP levels on the relationship between DCs and T cells.
The authors of the article must therefore take these data into account in order to avoid a one-sided presentation of the material that may mislead the readers of the article.
